# Effects of Crotonylation on Reprogramming of Cashmere Goat Somatic Cells with Different Differentiation Degrees

**DOI:** 10.3390/ani12202848

**Published:** 2022-10-19

**Authors:** Wennan Li, Wei Yan, Fei Hao, Lingyun Hao, Dongjun Liu

**Affiliations:** State Key Laboratory of Reproductive Regulation & Breeding of Grassland Livestock, Inner Mongolia University, Hohhot 010021, China

**Keywords:** somatic cells, lysine crotonylation, cloned embryo, epigenetic modification

## Abstract

**Simple Summary:**

Currently, not enough is known about the effect of histone modification on the epigenetic reprogramming of somatic cells, and the lack of basic study limits the development of somatic cell nuclear transfer technology. The aim of this study was to explore the influence of lysine crotonylation, a newly discovered histone post-translational modification, on the reprogramming of somatic cells from Cashmere goats. The results showed that the crotonylation level was increased in somatic cells with sodium crotonate treatment. At the same time, the treatment of somatic cells improved the cloned embryo cleavage rate. In conclusion, an increasing crotonylation level could promote the reprogramming of somatic cells and cloned embryo development. This finding provides an important reference for future improvements in the efficiency of in vitro Cashmere goat somatic cell nuclear transfer embryo production.

**Abstract:**

Failure in the epigenetic reprogramming of somatic cells is considered the main reason for lower cloned embryo development efficiency. Lysine crotonylation (Kcr) occupies an important position in epigenetic modification, while its effects on somatic cell reprogramming have not been reported. In this study, we detected the influence of sodium crotonate (NaCr) on the Kcr levels in three types of somatic cells (muscle-derived satellite cells, MDSCs; fetal fibroblast cells, FFCs; and ear tip fibroblast cells, EFCs). The three types of somatic cells were treated with NaCr for cloned embryo construction, and the cleavage rates and Kcr, H3K9cr, and H3K18cr levels in the cloned embryos were analyzed. The results showed that the abnormal levels of Kcr, H3K9cr, and H3K18cr were corrected in the treatment groups. Although there was no significant difference in the cloned embryo cleavage rate in the FFC treatment group, the cleavage rates of the cloned embryos in the MDSCs and EFCs treatment groups were increased. These findings demonstrated that the Kcr level was increased with NaCr treatment in somatic cells from Cashmere goat, which contributed to proper reprogramming. The reprogramming of somatic cells can be promoted and cloned embryo development can be improved through the treatment of somatic cells with NaCr.

## 1. Introduction

Somatic cell nuclear transfer (SCNT) technology holds broad application prospects in animal breeding, valuable germplasm conservation, and genetically engineered animal model construction. However, some problems (high miscarriage, deformity, and death rates and abnormal placental development) have resulted in low efficiency in recent years [1]. These factors have greatly limited the development and application of SCNT technology. Numerous studies have shown that the incomplete epigenetic reprogramming and epigenetic aberration of donor cells are the main reasons for the low efficiency of SCNT during cloned animal development [2,3]. Epigenetic modification issues, such as aberrant DNA hypermethylation, aberrant histone modification, and an abnormal inactive X chromosome, prevent zygotic genome activation (ZGA), ultimately giving rise to cloned embryo development failures [4,5].

Correcting epigenetic modifications on cloned embryos and somatic cells is of increasing topical interest. Histone deacetylase inhibitor (HDACi) exhibited a corrective effect on abnormal gene expression in cloned embryos [6,7]. In 2006, the Kishigami [8] and Rybouchkin [9] groups found that HDACi can increase the mouse embryo development rate from 1% to 6%. Moreover, the use of various types of HDACi (VPA [6], Scriptaid [10], and oxamflatin [11]) has also been found to facilitate pig cloned embryo development. In addition, the levels of H3K9me3 and H3K36me3 in cattle embryos were decreased by inducing the expression of Kdm4B in donor cells, leading to an improvement in reprogramming ability in cloned blastocysts [12]. The pretreatment of donor cells from sheep with recombinant Kdm4B can improve SCNT embryo development in vitro [13].

Lysine crotonylation (Kcr) is a newly discovered epigenetic modification that is widely present in most cellular histone and nonhistone proteins. Kcr is involved in the regulation of gene expression and spermatogenesis, the maintenance of stem cell pluripotency, the regulation of disease occurrence, and other functions [4]. Tian [14] and Ching [15] et al. treated ovarian granulosa cells with sodium crotonate (NaCr), leading to an improvement in the transition efficiency of ovarian granulosa cells into induced pluripotent stem cells and oocyte maturation in vitro. Kcr maintains the activation of sex-chromosome-linked genes in postmeiotic cells, an important modification in the gene reprogramming of postmeiotic cells [16]. Based on the above findings, Kcr is crucial to the epigenetic reprogramming of embryo development, while its influence on the reprogramming of donor cells from SCNT has not been reported.

The aim of this study was to understand the impact of Kcr on the epigenetic reprogramming of different types of donor cell from Cashmere goats. Muscle-derived satellite cells (MDSCs), fetal fibroblast cells (FFCs), and ear tip fibroblast cells (EFCs) were treated with sodium crotonate (NaCr), and the effects were analyzed in terms of cell cycle, apoptosis, pluripotency-associated gene expression, and Kcr level. These cells were used as donors to construct SCNT embryos to investigate the influence of histone Kcr on the reprogramming of somatic cells with different degrees of cell differentiation. This study offers a theoretical basis for further research on the roles of histone Kcr in somatic reprogramming and cloned embryo development.

## 2. Materials and Methods

### 2.1. Isolation and Culture of MDSCs

Briefly, skeletal muscle tissues were obtained from the hind limbs of an Arbas Cashmere goat female fetus at day 40 of gestation and washed with Dulbecco’s phosphate-buffered saline (DPBS) (BI, Kibbutz Beit HaemeK, Galilee, Israel) containing a 1% penicillin–streptomycin solution (BI, Israel). Skeletal muscle tissues were minced and digested with 0.1% collagenase I (sigmaaldrich, Burlington, MA, USA) for 1 h. The tissues were subsequently centrifuged to remove the supernatant and digested with 0.25% trypsin (BI, Kibbutz Beit HaemeK, Israel) for 20 min. Then, cells were filtered from the digestion mixture, centrifuged, seeded on a cell culture dish (Corning, NY, USA), and cultured in MDSC medium (DMEM/F12 medium containing 20% (*v/v*) fetal bovine serum (FBS) and 10% horse serum (HS)) at 37 °C with 5% CO_2_. MDSCs were subcultured when confluence reached 70%. The MDSC medium was used for all MDSC culture experiments.

### 2.2. FFC and EFC Cultures

FFCs and EFCs were previously reserved from our lab. FFCs were derived from a female 40-day-old Arbas Cashmere goat fetus. EFCs were obtained from a 1-year-old female Arbas Cashmere goat. The 4th-passage FFCs and EFCs were rapidly thawed in a 37 °C water bath and cultured in a DMEM/F12 medium containing 10% FBS in culture dishes. All cultures were maintained at 37 °C with 5% CO_2_ in a cell incubator (Thermo Fisher Scientific, Waltham, MA, USA), and the medium was replaced every other day. FFCs and EFCs were passaged when 70% confluence was reached. The medium of FFCs and EFCs was used for all FFC and EFC culture experiments.

### 2.3. Cell Immunofluorescence 

The 5th-passage MDSCs, FFCs, and EFCs (1 × 10^4^ cells/well) were cultured onto coverslips and fixed with 4% paraformaldehyde for 30 min. Fixed cells were permeabilized with 0.1% Triton X-100 for 15 min on ice. Then, cells were blocked in 5% bovine serum album for 1h at 37 °C and incubated with primary antibodies overnight at 4 °C. This was followed by incubation with secondary antibodies (Donkey Anti-Rabbit IgG H+L (Alexa Fluor^®^ 488) (Abcam, Carlsbad, CA, USA) for 1h in the dark. The cells were incubated with DAPI for 20 min to stain nuclei, and cell immunofluorescence images were acquired using a Nikon microscope (Nikon-Air, Nikon instruments Co., LTD, Tokyo, Japan). Negative controls were incubated with PBS instead of the primary antibody. See Appendix A for a list of the primary antibodies used. 

### 2.4. Growth Curve of MDSCs, FFCs, and EFCs

The 5th-passage MDSCs, FFCs, and EFCs (2 × 10^4^/per well) were seeded into 24-well plates and cultured at 37 °C with 5% CO_2_. Three wells from each of the MDSCs, FFCs, and EFCs were trypsinized every 24 h, counted with a hemocytometer, and averaged to determine the number of cells per well. The lines were recorded for 7 days to generate growth curves.

### 2.5. Western Blot

The 5th-passage MDSCs, FFCs, and EFCs (1 × 10^4^ cells/well) were respectively cultured to 70% confluency at 37 °C with 5% CO_2_ and collected for a Western blot analysis. In brief, total protein and histone protein from MDSCs, FFCs, and EFCs were extracted by a Mammalian Protein Extraction Kit (CWBIO, Jiangsu, China) and an EpiQuik Total Histone Extraction Kit (EpiGentek, Farmingdale, NY, USA), according to the instructions. Total protein and histone protein were quantified using a BCA Protein Assay Kit (Thermo, Carlsbad, CA, USA). The protein samples were separated on 12% SDS–PAGE gels and transferred onto a nitrocellulose membrane (Millipore, Burlington, MA, USA). The membranes were blocked by 5% skimmed milk for 1 h and subsequently incubated with primary antibodies at 4 °C overnight. Then, the membrane was washed with TBST and incubated with HRP secondary antibodies (Proteintech, Chicago, IL, USA, diluted 1:10,000) for 1 h at 25 ± 2 °C. The protein bands were visualized using ECL Western Blotting Substrate (Thermo Fisher Scientific, Carlsbad, CA, USA). The relative intensity of the bands was quantified by ImageJ software (Image J1.5 software, NIH, Bethesda, MD, USA). The concentrations and manufacturers of the primary antibodies are listed in Appendix A.

### 2.6. Cell Viability Assay

NaCr is a yellow-green powder (391719, Sigma, Darmstadt, Germany). A concentrated stock solution of NaCr was prepared by dissolving in DMEM/F12 or ultrapure water to a concentration of 25 mg/mL. The NaCr solution was added to the culture medium on demand. Cell viability was measured by Cell Counting Kit–8 (CCK–8, Yesen, Shanghai, China) according to the manufacturer’s instructions. The 5th-passage MDSCs, FFCs, and EFCs were seeded into 96-well plates at a density of 2  ×  10^3^ cells/well. After adherence for 24 h, each cell line was treated with NaCr at different concentrations (0 mM, 10 mM, 20 mM, 30 mM, and 40 mM) for 24 h, 48 h, and 72 h. After the NaCr treatment, a CCK–8 solution was added to each well and incubated for 1 h. Then, the absorbance was analyzed at 450 nm by a microplate reader (Varioskan Flash, Thermo Fisher Scientific, Waltham, MA, USA). All experiments were performed three times. All cultures were maintained at 37 °C with 5% CO_2_. 

### 2.7. Screening Optimal Incubation Time and NaCr Concentration

To determine the optimal incubation time, the 5th-passage MDSCs, FFCs, and EFCs grown to 60% confluency were treated with NaCr (10 mM) for different times (12 h, 24 h, 48 h, and 72 h) at 37 °C with 5% CO_2_. The levels of Kcr and lysine acetylation (Kac) were detected by Western blot after incubation. After the optimal incubation time was identified, different incubation concentrations (10 mM, 20 mM, 30 mM, and 40 mM) were tested to select the appropriate concentration of NaCr for future experiments. The levels of Kcr, H3K9cr, H3K18cr, and Lamin B1 were detected by Western blot after NaCr treatment. The Western blot procedure was similar to that described in Section 2.5. The antibodies used and concentrations are detailed in Appendix A.

### 2.8. Cell Cycle and Apoptosis Analysis

Briefly, the 5th-passage MDSCs, FFCs, and EFCs were cultured to 60% confluence and treated with NaCr at different concentrations (10 mM, 20 mM, 30 mM, and 40 mM) for 24 h, 48 h, and 72h. The negative control group was performed by replacing the NaCr treatment with normal saline. Cells in the control group were treated without NaCr and cultured with normal medium. All treated cells were collected and washed with cold PBS for use in the subsequent cell apoptosis analysis. 

For the cell apoptosis analysis, an Annexin V-FITC/PI Apoptosis Detection Kit (7sea biotech, China) was used, following the manufacturer’s protocol. The treated and collected cells were incubated with the Annexin V-FITC solution for 15 min and with PI for an additional 5 min in the dark. 

A cell cycle analysis was performed via a Cell Cycle and Apoptosis Analysis Kit (7sea biotech, China). Each cell was fixed with cold 70% ethanol at 4 °C overnight and then stained with a PI solution for 30 min at 37 °C. 

The cell cycle and apoptosis were analyzed by CytoFLEX S flow cytometry (Beckman, USA). All experiments group were performed three times.

### 2.9. Quantitative Real-Time PCR (qRT-PCR) 

Briefly, total RNA was extracted from each cell sample receiving the NaCr treatment (treatment condition in Section 2.8) using RNAiso Plus Reagent (Takara, Kyoto, Japan) and reverse-transcribed using the PrimeScript RT-PCR kit with gDNA Eraser (Takara, Kyoto, Japan). The cDNA templates were tested for the absence of contaminated genomic DNA using *GAPDH* primers that amplify different fragment sizes with genomic DNA and cDNA templates (Appendix A). The obtained cDNA was used as a template for qRT–PCR. qRT-PCR reactions were performed with TB Green Premix Ex Taq II (Til RNaseH Plus, Takara, Kyoto, Japan) and run on a CFX96 (Bio-Rad) system. The primers used are listed in Table 1. The relative mRNA levels were analyzed using the 2^−ΔΔCt^ method (normalized with *GAPDH*).

### 2.10. Oocyte Collection and IVM

Fresh ovaries of Arbas Cashmere goats were obtained and washed with saline. The washed ovaries were placed in the dish containing the collecting oocyte solution. Follicles from ovaries were then cut with a scalpel, allowing the release of cumulus–oocyte complexes (COCs). The COCs were transferred to a maturation medium (TCM199 medium containing 10% estrous goat serum [17], 0.1 μg/mL β-estradiol (Wako, Osaka, Japan), 10 μg/mL FSH (Sansheng Pharmaceutical Co. Ltd., Zhejiang, China), 8 μg/mL LH (Sansheng Pharmaceutical Co. Ltd., Zhejiang, China), 10 mM HEPES (Sigma, Darmstadt, Germany), 2 mM sodium pyruvate (Wako, Osaka, Janpan), and a 1% penicillin–streptomycin solution (BI, Kibbutz Beit HaemeK, Israel)) for 22 h at 38 °C.

### 2.11. IVF 

Fresh sperm from Arbas Cashmere goats was diluted and incubated for 30 s to 2 min at 37 °C using Bo-SemenPrep (IVF BioScience, Cornwall, UK), following the manufacturer’s instructions, and then capacitated sperm were collected from the suspension semen. The maturation COCs obtained from the previous step were washed twice in an IVF fertilization medium and transferred to drops of IVF fertilization medium (BO-IVF- fertilization medium, IVF Bioscience, UK). At the same time, capacitated sperm was added into and incubated with COCs for 6 h at 38.5 °C. Fertilized eggs were then transferred to drops of IVC culture medium (BO-IVC-culture medium, IVF Bioscience, England). The cumulus cells from fertilized eggs were removed by mechanical pipetting and washed three times with IVC culture medium. Finally, the fertilized eggs were cultured in drops of IVC culture medium, and cleavage was observed under a microscope after 48h.

### 2.12. SCNT 

The 5th-passage MDSCs, FFCs, and EFCs were cultured to 60% confluence and treated with the indicated concentration of NaCr at 37 °C with 5% CO_2_. These cells were used as donor cells. After IVM, the cumulus and granulosa cells of maturation COCs were removed by hyaluronidase (Sigma, Saint Louis, MO, USA) and transferred to drops of cytochalasin B (Sigma, Saint Louis, MO, USA). The oocytes were enucleated with micromanipulation by the aspiration of the first polar body and part of the adjacent cytoplasm. The NaCr-treated donor cells (MDSCs, FFCs, and EFCs) were injected into the enucleated oocyte and fused by two 180V/mm direct current pulses for 20 μs. The fused cloned embryos were transferred into IVM medium and incubated for 30 min. Then, reconstructed embryos were activated with 5 μM calcium ionophore A23187 (Sigma, Darmstadt, Germany) for 5 min, followed by incubation with synthetic oviduct fluid (SOFaa) containing 2 mM 6-dimerhylaminepurine (6-DMAP) (Sigma, Darmstadt, Germany) for 3.5 h to complete the activate process. After activation, cloned embryos were washed with IVC culture medium and transferred to fresh IVC medium to culture at 38 °C with 5% CO_2_. Cleavage was observed after 48 h.

### 2.13. Immunofluorescence Analysis of Embryo

Embryos were collected from IVF and SCNT and fixed in 4% PFA/PBS (Solarbio, Beijing, China) for 30 min (light was avoided). Following fixation, embryos were washed with washing buffer and permeabilized by 1% Triton overnight at 37 °C. Afterwards, embryos were blocked in 2% BSA for 1 h and incubated with the primary antibody for approximately 12 h at 4 °C. Counterstaining with the secondary antibody (Donkey Anti-Rabbit IgG H+L (Alexa Fluor^®^ 488)) and DAPI (Solarbio, China) was performed as described in Section 2.3. The immunofluorescence results were detected under a Nikon microscope (Nikon-Air, Nikon Instruments Co., LTD, Japan). The antibodies used and the manufacturers of the primary antibodies are listed in Appendix A. 

### 2.14. Statistical Analysis

Each experiment was replicated three times, and statistical analysis was performed with Student’s *t*-test using GraphPad Prism 8 (GraphPad 8.0.1 software, San Diego, CA, USA). The results are presented as means ± standard deviations of the mean. A *p*-value < 0.05 was considered statistically significant.

## 3. Results

### 3.1. Somatic Cell Preparation and Culture

The isolated MDSCs were characterized by immunofluorescence for the surface markers PAX7, Myf5, and MyoD1. The results revealed the MDSC markers were positive (Figure 1a). The MDSCs, EFCs, and FFCs exhibited a spindle-like morphology, and the growth curves were S-shaped (Figure 1b–c). The growth of EFCs was slow among the three types of somatic cells (Figure 1c).

### 3.2. Detection of Kcr Level and of Pluripotency Gene Expression in Somatic Cells

The immunofluorescence results revealed that the presence of Kcr modifications and the fluorescence intensity of Kcr-positive cells were different among the three types of somatic cells (Figure 1d). There was no significant difference in fluorescence intensity in the FFC and EFC groups, and the MDSC group displayed the highest fluorescence intensity among the three groups (Figure 1d). These results were further confirmed by the Western blot analysis. The Kcr levels differed among the three types of somatic cells and were highest in the MDSC group (Figure 1e). However, no significant differences in the cytoplasmic protein Kcr level were observed between the three types of somatic cells (Figure 1f).

In addition to the assessment of the Kcr level, the expression of pluripotency factor *Oct4*, *Sox2*, and *Nanog* was compared between the three types of cells (MDSCs, FFCs, EFCs) by Western blot analysis. As shown in Figure 1g, *Nanog* expression was very low in the FFC and EFC groups. The *Oct4* expression in FFCs was lower than that in EFCs and MDSCs, whereas *Sox2* expression was higher in FFCs than in EFCs and MDSCs. 

### 3.3. Effect of NaCr on Cell Viability and Screening of Optimal Incubation Time and Concentration

To detect the effects of NaCr on the cell viability of MDSCs, FFCs, and EFCs, these cells were treated with gradient concentrations of NaCr for various times. As shown in Figure 2a, the cell viability of the MDSC and FFC groups under the same NaCr concentration followed an uptrend as time progressed. However, the cell viability of MDSCs and EFCs decreased with increasing NaCr concentration at the same time point (Figure 2a). As shown in Figure 2a, 10 mM NaCr treatments of MDSCs, FFCs, and EFCs at the 24 h, 48 h, and 72 h time points demonstrated the highest cell viability.

Based on the obtained cell viability results, the three cells were treated with 10 mM NaCr, and the Kcr level was detected by a Western blot analysis. The Kcr level was upregulated in treated MDSCs, FFCs, and EFCs at different time points (Figure 2b–d). By contrast, there was no significant difference in the Kac level of treated MDSCs and EFCs at 24 h compared with the MDSC and EFC control groups, respectively (Figure 2b,d). Although a decrease in the Kac level of FFCs was observed after 24 h of treatment, a high level of Kcr was detected in FFCs after 24 h of Nacr treatment (Figure 2c). This result indicates that, in the case of no change in Kac, 24 h of treatment was the most appropriate time for selection.

To further confirm the appropriate concentration, the three types of cells were treated with a concentration gradient of NaCr for 24 h, and the levels of H3K9cr and H3K18cr in the cells were analyzed by Western blot. The H3K18cr and H3K9cr levels clearly increased in the MDSC, FFC, and EFC treatment groups compared to the respective control groups (Figure 2e–g). The H3K9cr level was significantly higher in the MDSCs treated with 30 mM NaCr, and the H3K18cr level also was clearly elevated in MDSCs after 10 mM, 20 mM, and 30 mM treatments (Figure 2e). Considered together, the most appropriate concentration of NaCr was 30 mM for MDSCs. As shown in Figure 2f, compared with the control groups, the H3K9cr and H3K18cr levels were significantly increased in the 20 mM NaCr treatment of the FFC group. Furthermore, there was a notable increase in the H3K9cr and H3K18cr levels with the 20 mM treatment in the EFC group compared to the other groups (Figure 2g). Thus, an NaCr concentration of 20 mM seemed to be appropriate for FFCs and EFCs. 

### 3.4. Effect of Optimal Incubation Time and NaCr Concentration on Apoptosis and Cell Cycle in Somatic Cells

Our results revealed that the percentage of apoptotic cells increased with NaCr concentration and treatment time (all results shown in Appendix A). We focused on the apoptosis rates of MDSCs, FFCs, and EFCs with 10 mM, 20 mM, 30 mM, and 40 mM NaCr treatments for 24 h, which contributed to further verifying the effect of the optimal NaCr concentration on cell apoptosis. As shown in Figure 3a, the proportion of apoptotic cells decreased in 30 mM treatment MDSC group compared with 40 mM treatment MDSC group. At the same time, the proportion was not significantly different between the 30 mM and 20 mM treatment MDSC groups. Therefore, 30 mM and 24 h NaCr are appropriate treatment conditions for MDSCs. There were no significant differences among the 10 mM, 20 mM, and 30 mM treatment FFC groups (Figure 3a). In the EFC group, the percentage of apoptotic cells was remarkably lower in the 20 mM group than in the 30 mM group, and the percentage did not significantly differ between the 20 mM and 10 mM groups (Figure 3a). Therefore, 20 mM and 24 h NaCr are suitable treatment conditions for FFCs and EFCs. 

The effect of the NaCr treatment on the cell cycle assay in MDSCs, FFCs, and EFCs was similar to that of the apoptosis assay. Increasing the treatment time and concentration resulted in the progressive inhibition of the cell cycles of the three types of somatic cells (all results shown in Appendix A). As shown in Figure 3b, the proportion of cells at the G0/G1 phase and the G2/M phase was not markedly different in the 30 mM treatment MDSC group compared with 20 mM and 40 mM treatment MDSC groups. In the FFC group, no significant difference in the proportion of cells in the G0/G1 phase and the G2/M phase was observed in the 20 mM treatment group compared with the 30 mM treatment group. The 20 mM treatment group had a higher proportion of cells in G0/G1 phase and a lower proportion of cells in G2/M phase than the 10 mM treatment group (Figure 3b). No significant changes were identified in the proportion of cells in the G0/G1 phase and G2/M phase among the 10 mM, 20 mM, and 30 mM treatment EFC groups (Figure 3b).

### 3.5. Effect of NaCr on the Expression of Key Pluripotency Genes in Somatic Cells

To determine whether NaCr affects the expression of pluripotency genes in somatic cells, three types of somatic cells were treated with various concentrations of NaCr for different incubation times, and the mRNA levels of *Oct4*, *Sox2*, and *Nanog* were analyzed by qRT-PCR. As shown in Figure 4a, *Nanog* expression was suppressed in the 30 mM 24 h treatment MDSC group. The expression of *Sox2* and *Oct4* increased, but this was nonsignificant in the same treatment MDSC group. When FFCs were treated with 20 mM NaCr for 24 h, the expression levels of *Oct4* and *Sox2* in FFCs were not decreased and *Nanog* expression level was increased (Figure 4b). At the same time, *Sox2*, *Oct4*, and *Nanog* expression were increased in the 20 mM/24 h treatment EFC group (Figure 4c). These results demonstrated that the NaCr concentration and the treatment time did not appreciably affect the expression of *Sox2* and *Oct4* in MDSCs and FFCs but did affect the expression of *Nanog* in MDSCs and FFCs. Treatment with 20 mM/24 h NaCr can promote the expression of pluripotency genes in EFCs. 

### 3.6. Comparison of Histone Kcr Modification Levels between IVF Embryos and Cloned Embryos

Treated (30 mM NaCr) and untreated MDSCs, respectively, served as two types of donor cells for constructing the SCNT embryo experimental (T-SCNT) and control (C-SCNT) groups. Following the same approach, FFCs and EFCs with 20 mM NaCr, respectively, served as donor cells for constructing the SCNT embryo experimental groups (FFC T-SCNT and EFC T-SCNT). The untreated FFCs and EFCs were used to construct control SCNT embryo groups (FFC C-SCNT and EFC C-SCNT).

The effects of treated donor cells (MDSCs) on the histone Kcr levels in SCNT embryos at different stages are analyzed in Figure 5a,b. The Kcr levels were clearly higher in the IVF group at different stages than in the MDSC SCNT embryo groups (C-SCNT and T-SCNT) (Figure 5a). The Kcr levels in the C-SCNT groups were significantly lower at the 1-cell, 4-cell, and 8-cell embryo stages than in the T-SCNT groups (Figure 5a,b). This result illustrates that MDSCs treated with NaCr, which served as donor cells, led to Kcr level increases in the SCNT embryos. To further clarify the effect of the NaCr treatment of somatic cells (MDSCs) on the development of SCNT embryos, the cleavage rates in the IVF, T-SCNT, and C-SCNT groups in MDSCs were measured. As shown in Appendix A, the numbers of embryos produced in the C-SCNT and T-SCNT groups were lower than that in the IVF group at each developmental stage. The cleavage rates in the T-SCNT and C-SCNT groups were significantly lower than that in the IVF group (Figure 5c). Moreover, the cleavage rate in the MDSC T-SCNT group was higher than that in the MDSC C-SCNT group (Figure 5c). 

To analyze the effects of treated donor cells (FFCs) on the Kcr modification of SCNT embryos at different stages, the H3K9cr levels were compared in the IVF and SCNT embryo (T-SCNT and C-SCNT) groups (Figure 6a,b). A lower level of H3K9cr was observed in the C-SCNT group at the 1-cell, 2-cell, 4-cell, and 8-cell embryos stages than that in the IVF group (Figure 6a). When donor cells (FFCs) were simultaneously treated with NaCr, the H3K9cr level was found to be significantly higher in the T-SCNT group at the 1-cell, 4-cell, and 8-cell stages compared to the C-SCNT group (Figure 6b). Appendix A presents the number of developmental embryos for IVF, T-SCNT, and C-SCNT at the two-to eight-cell stage. The number of embryos from each embryonic stage in the IVF group was higher than in T-SCNT and C-SCNT. However, there was no significant difference in the cleavage rate between the T-SCNT and C-SCNT groups (Figure 6c). 

In order to study the influence of treated donor cells (EFCs) on the Kcr modification of SCNT embryos at different stages, H3K18cr levels were detected in the IVF and SCNT embryo (T-SCNT and C-SCNT) groups (Figure 7a,b). The EFC C-SCNT group showed a significantly lower level of H3K18cr at the 1-cell, 2-cell, 4-cell, and 8-cell stages than the IVF group (Figure 7a). In addition, our findings revealed that the H3K18cr level was markedly elevated in the T-SCNT group at the 2-cell and 4-cell stages compared to the C-SCNT group (Figure 7b). The counted embryo numbers shown in Appendix A is from two-to-eight-cell stage for each group. This result demonstrates that the T-SCNT group had a higher cleavage rate compared with the C-SCNT group (Figure 7c and Appendix A). 

## 4. Discussion

A variety of somatic cells were used as donor cells in the development of SCNT technology. The previous study showed that different types of donor cells could affect the efficiency of SCNT [18]. Researchers compared the developmental potential of SCNT embryos using granulosa cells, colostrum-derived mammary gland epithelial cells, and ear-derived fibroblast cells as nuclei donor. The highest development rates of blastocysts were achieved using granulosa cells as donor cells, and six cloned cattle were born healthy [19]. Colostrum-derived mammary gland epithelial cells produce a better full-term developmental embryo by nuclear transfer [20]. This may suggest the contribution of different levels of epigenetic modification in various donor cells [21]. Researchers clearly pointed out that low DNA methylation levels or high acetylation levels from donor cells facilitate reprogramming [22]. Therefore, we selected three different types of somatic cells to achieve a comprehensive understanding of the effects of Kcr on the reprogramming of somatic cells. We confirmed that there was no difference in the cytoplasmic Kcr level, but there was a significant difference in the nuclear histone Kcr level among the three types of somatic cells. Previous research revealed that pluripotency genes maintain pluripotency and regulate the differentiation of stem cells [23]. The rates and directions of cells differentiation were determined by the transcription factor *Sox2* [24]. Aberrant expression of *Sox2* can lead to a failure to maintain the homeostasis of cell differentiation [25]. *Nanog* is essential for early embryonic development. *Nanog* expression decreases gradually during the differentiation of embryonic stem cells and is weak or disappears after birth [26]. Pluripotency factor deletion in stem cells can lead them to lose pluripotency and progressively become terminally differentiated cells. In our study, the different expression levels of pluripotency genes in each somatic cell were determined, which may reflect the different degrees of differentiation of each somatic cell. In addition, previous findings suggest that terminally differentiated cells are more refractory to reprogramming in SCNT versus low-differentiated cells [27]. This may be one of the reasons why reprogramming ability differs among different somatic cells. 

Different histone modification sites yield different reprogramming efficiencies in somatic cells [28]. In 2009, Yamanaka [29] proposed that the Kac level in the donor cell genome appeared to be positively correlated with the development rate of SCNT embryos of miniature pigs. Previous studies showed that Kac modification was enhanced by adding HDACi to the somatic cells to improve the developmental competence of mouse SCNT embryos [30]. In our study, when MDSCs, FFCs, and EFCs were treated with 10 mM NaCr for 24 h, the Kac levels of the three types of somatic cells were not markedly decreased or increased. In order to avoid the influence of increasing Kcr levels in our experiment, we selected 24 h as the incubation time for the subsequent experiments.

The function and structure of Kcr and Kac are similar, and their modification sites are also partially overlapping [31,32]. The Kcr target sites H3K18 and H3K9, two main sites of Kcr modification, have been conserved throughout evolution [33]. There is significant abundance at the Kcr level of these modification sites (H3K18 and H3K9) compared with Kac [33]. Our study showed that only 30 mM NaCr can enhance the levels of both H3K9cr and H3K18cr in MDSCs. Likewise, FFCs and EFCs treated with 20 mM NaCr can achieve similar histone modification effects. The optimal NaCr concentration and treatment time did not give rise to an excessively high rate of apoptosis at the same time. Based on these results, the optimal NaCr treatment concentration and incubation times were determined. In addition, previous studies found that the crotonylated proteins were mainly enriched in biological processes such as RNA processing, nucleic acid metabolic processes, and chromosome organization [34]. This implicated crotonylated proteins as being involved in and important for RNA and DNA synthesis. Studies also further demonstrated that an excessively high concentration of Kcr would inhibit the cell cycle and proliferation [35]; this phenomenon was also observed in our study. We found the number of donor cells in the G0/G1 phase increased as the NaCr concentration increased. This result is consistent with what has been described in a previous study. NaCr appears to inhibit cell cycle progression, with more cells arrested in the G0/G1 phase. Other studies found that somatic cells in the G0/G1 phase are prone to undergo cell reprogramming and are considered to be ideal donor cells [36]. In our follow-up SCNT experiment, cell cycle arrest may have positively affected SCNT. In recent years, research has shown that the pluripotency genes *Nanog*, *Sox2*, and *Oct4* play key roles in the reprogramming process of somatic cells [18]. The activity of pluripotency transcription factors was modulated through different histone modifications, and activated factors are involved in the regulatory processes of reprogramming in somatic cells [37]. Our results demonstrated that different concentrations of NaCr give rise to changes not only in the Kcr level but also in the pluripotency gene expression level at the different treatment time points. Unfortunately, the change does not occur in a NaCr-dose-dependent manner. Based on this, we speculate that activated pluripotency transcription factors were not directly modulated through histone crotonylation modification. This hypothesis needs further exploration in the future. 

Abnormal or incomplete nuclear reprogramming results in a low rate of cloned embryo development compared to IVF embryos [38]. In our study, we found that the Kcr, H3K18cr, and H3K9cr levels in IVF embryos were significantly higher at different stages. This suggests that Kcr modification is involved in the epigenetic reprogramming of somatic cells. However, the Kcr, H3K18cr, and H3K9cr levels were lower in SCNT embryos. This result also demonstrated that, unlike the process in IVF embryos, epigenetic reprogramming in somatic cells is incomplete, leading to abnormal development in SCNT embryos. The researchers attempted to implement some repair means to correct the reprogramming of somatic cells and the aberrant development of cloned embryos [39,40]. Previous studies have found that somatic cell reprogramming and the developmental competence of cloned embryos were improved by correcting epigenetic modifications [41]. For example, a higher proportion of embryo development occurs at the 8-cell-to-blastocyst stage during SCNT by changing the methylated DNA level in somatic cells [42]. In our experiment, after the somatic cells were treated with NaCr, the Kcr, H3K18cr, and H3K9cr levels were increased in somatic cells and cloned embryos. The histone modification levels of Kcr, H3K9cr, and H3K18cr in cloned embryos were similar to those in the IVF group. At the same time, the developmental competence of the cloned embryos was improved. We speculate that Kcr loosens the chromatin’s higher-order structure to allow certain target transcription factors bind to the newly available sites and promotes transcription factor activation [43]. The activation of transcription leads to the successful reprogramming of somatic cells and embryo development [44]. 

Interestingly, we found that, although the cleavage rate of cloned embryos in FFC T-SCNT group was not significantly different compared with that of the FFC C-SCNT group, the H3k9cr level was still increased in the T-SCNT group. Since an improvement in the development of cloned embryos may require the synergism of multiple forms of epigenetic modification, a single type of epigenetic modification is unlikely to completely correct the reprogramming of all types of somatic cells during SCNT. Each epigenetic modification may be involved in the generation of different reprogramming results of different donor cell lines with different initial genetic backgrounds [45]. For example, EFCs treated with HDACi could improve the blastocyst development of bovine SCNT embryos, but the same treatment in bone marrow cells did not affect embryo development [46]. Therefore, we are aware of the limitations of the current knowledge regarding the effect of epigenetic modification on the development of cloned embryos and animals. These limitations highlight the need for further research in this field.

## 5. Conclusions

In summary, this study demonstrated that somatic cells treated with NaCr could improve SCNT embryonic development. Furthermore, enhancing the Kcr levels in somatic cells contributed to the proper reprogramming of three types of somatic cells from Cashmere goats. The present findings may provide insights relevant to the effect of Kcr on SCNT embryo development.

## Figures and Tables

**Figure 1 animals-12-02848-f001:**
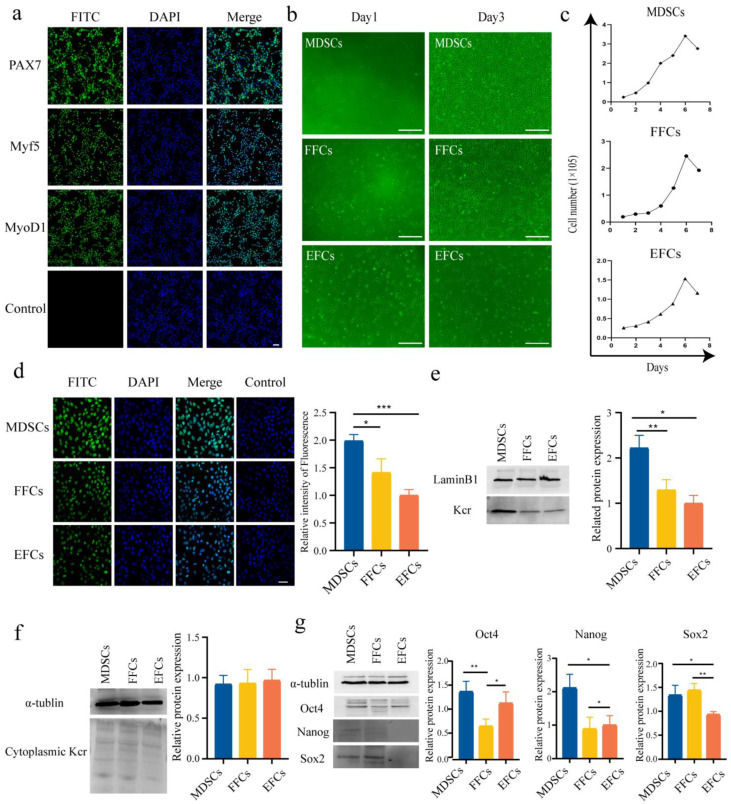
Preparation and characterization of three types of somatic cells. (**a**) The surface markers PAX7, Myf5, and MyoD1 of MDSCs were evaluated by immunofluorescence. PBS was used instead of the primary antibody as a negative control; nuclei (blue), target proteins (green) (scale bar, 100 µm); muscle-derived satellite cells: MDSCs; fetal fibroblasts cells: FFCs: ear tip fibroblasts cells: EFCs. (**b**) Three types of cultured somatic cells photographed on day 1 and day 3 (Scale bar, 500 µm). (**c**) The growth curves of three types of somatic cells. (**d**) Kcr levels in three types of somatic cells were examined and quantified by immunofluorescence; the negative control was prepared using PBS instead of the primary antibody (scale bar, 50 µm); Kcr: lysine crotonylation. (**e**) Western blotting for levels of histone Kcr in three types of somatic cells. Image J was used to quantify grey value. (**f**) Kcr level was examined by Western blot in cytoplasm, and a quantitative analysis of grey value was performed. (**g**) The expression of *Oct4*, *Sox2*, and *Nanog* was detected by Western blot, and the relative protein expression levels were quantified according to the protein grey value; * *p* < 0.05, ** *p* < 0.01, *** *p* < 0.001, no asterisk indicates no significant difference.

**Figure 2 animals-12-02848-f002:**
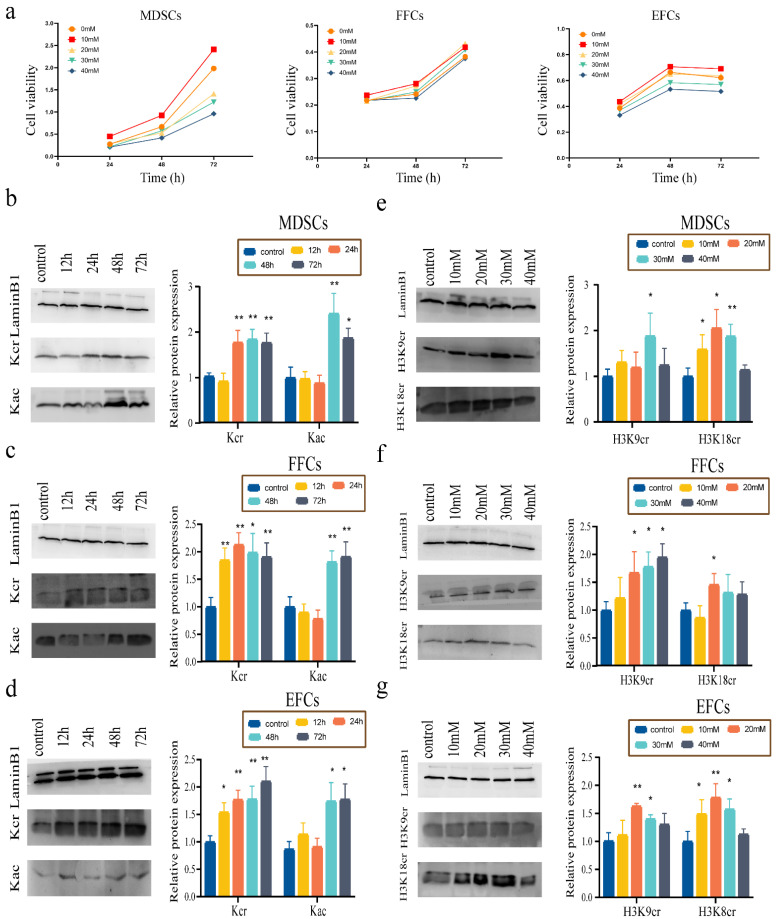
Effect of NaCr on the cell viability and screening the optimal incubation time and concentration. (**a**) Cell viability assessment of treated MDSCs, FFCs, and EFCs with various NaCr concentrations and testing times. (**b**–**d**) Western blotting and relative quantification of the Kcr and Kac levels in MDSC, FFCs, and EFCs with 10 mM NaCr treatments at various time points. (**e**–**g**) Western blotting and relative quantification of the H3K9cr and H3K18cr levels in MDSC, FFCs, and EFCs treated with an NaCr concentration gradient for 24 h; * *p* < 0.05, ** *p* < 0.01; no asterisk indicates no significant difference; control group: cells that received no treatment.

**Figure 3 animals-12-02848-f003:**
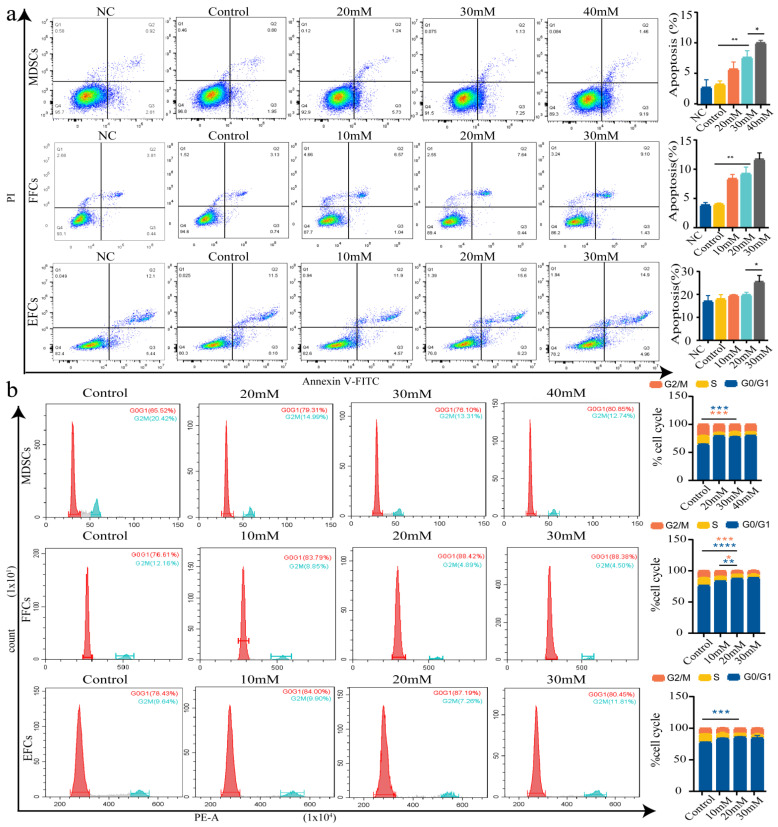
Apoptosis and cell cycle of somatic cells treated with different concentrations of NaCr. (**a**) Apoptosis assessment of MDSCs, FFCs, and EFCs treated with 10 mM, 20 mM, 30 mM, and 40 mM NaCr for 24 h by flow cytometry. The histogram shows the apoptosis statistics. Control group: cells that received no treatment. Negative control group (NC): cells that were treated with normal saline. (**b**) A cell cycle analysis was used to detect the cell cycle distribution after the NaCr treatments. Blue asterisks represent significant differences in the G0/G1 phase among the different groups. Pink asterisks indicate significant differences in the G2/M phase among the different groups. Control group: cells that received no treatment. * *p* < 0.05, ** *p* < 0.01, *** *p* < 0.001, **** *p* < 0.0001; no asterisk indicates no significant difference.

**Figure 4 animals-12-02848-f004:**
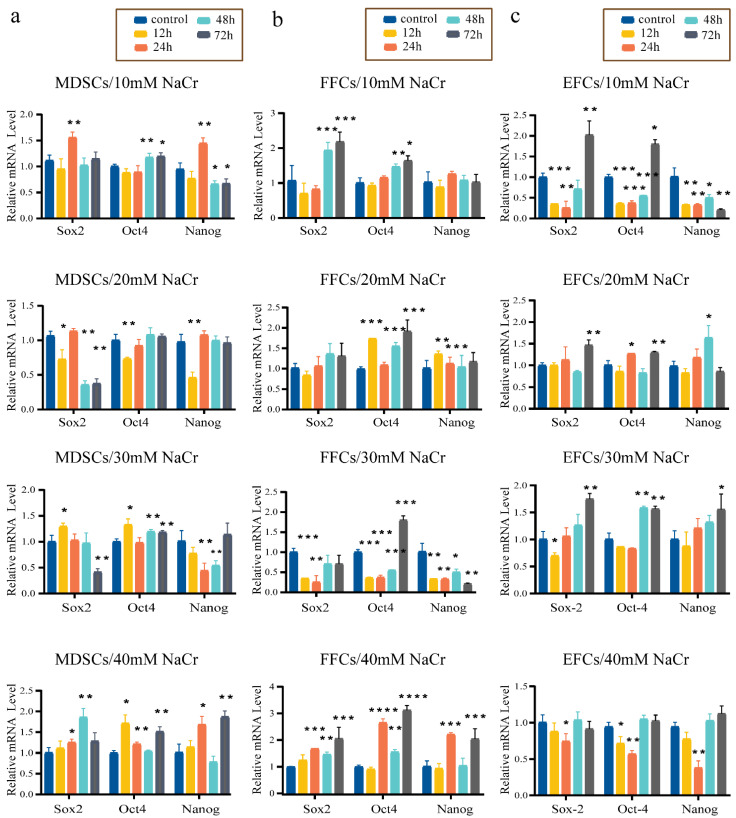
Expression of key pluripotency genes of MDSCs, FFCs, and EFCs treated with different NaCr concentrations for incubation times. (**a**–**c**) mRNA levels of *Sox2*, *Oct4*, and *Nanog* in MDSCs, FFCs, and EFCs after various Nacr treatment concentrations at different time points; untreated MDSCs, FFCs, and EFCs served as the (**a**) control, (**b**) control, and (**c**) control, respectively; * *p* < 0.05, ** *p* < 0.01, *** *p* < 0.001, **** *p* < 0.0001 vs. control; no asterisk indicates no significant difference.

**Figure 5 animals-12-02848-f005:**
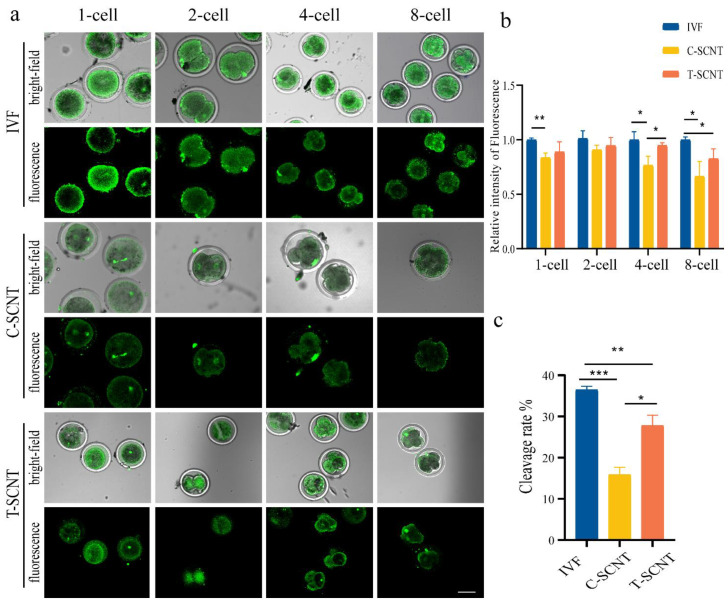
Effect of treatment of donor cells (MDSCs) with NaCr on the level of Kcr and developmental competence in SCNT embryos. (**a**,**b**) Fluorescence staining pattern and relative fluorescence intensity of Kcr level in IVF and MDSC SCNT embryos (C-SCNT and T-SCNT) at different stages (scale bar, 50 µm). (**c**) The cleavage ratio of embryos at all cleavage stages in IVF, C-SCNT, and T-SCNT. the C-SCNT (control SCNT) and T-SCNT (experimental SCNT) group represent cloned embryos generated with untreated and NaCr-treated donor cells (MDSCs), respectively. IVF group represents blank control group; * *p* < 0.05, ** *p* < 0.01, *** *p* < 0.001; no asterisk indicates no significant difference.

**Figure 6 animals-12-02848-f006:**
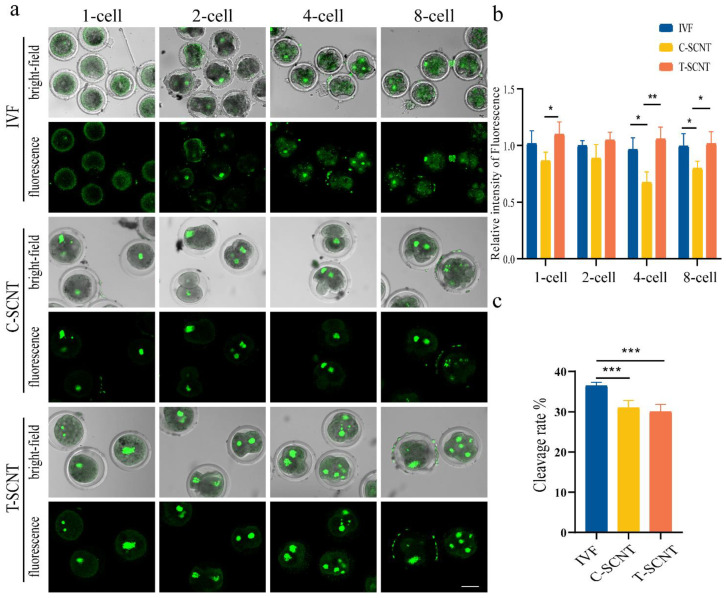
Effect of treatment of donor cells (FFCs) with NaCr on the level of H3K9cr and the developmental competence in SCNT embryos. (**a**,**b**) Fluorescence staining pattern and relative fluorescence intensity of the H3K9cr level in IVF and FFC SCNT embryos (C-SCNT and T-SCNT) at different stages (scale bar, 50 µm). (**c**) The cleavage ratios of embryos at all cleavage stages in IVF, C-SCNT, and T-SCNT. The C-SCNT (control SCNT) and T-SCNT (experimental SCNT) groups represent cloned embryos generated with untreated and NaCr-treated donor cells (FFCs), respectively. The IVF group represents a blank control group; * *p* < 0.05, ** *p* < 0.01, *** *p* < 0.001; no asterisk indicates no significant difference.

**Figure 7 animals-12-02848-f007:**
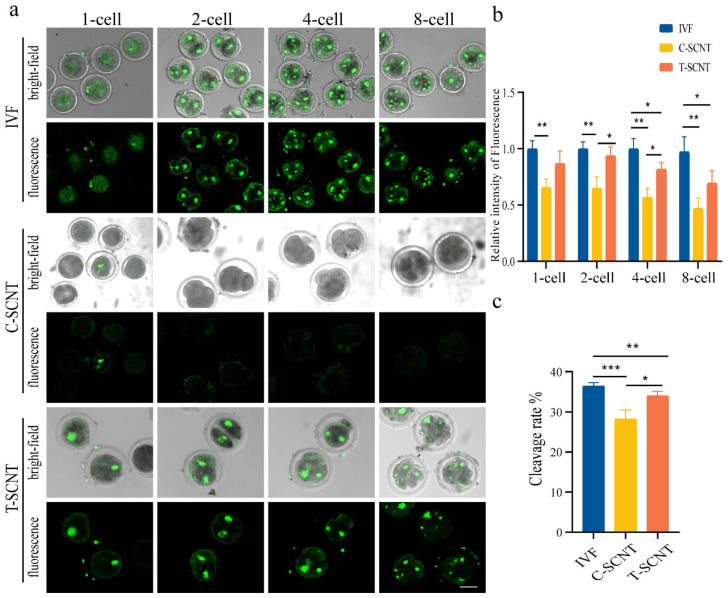
Effect of treatment of donor cells (EFCs) with NaCr on the level of H3K18cr and developmental competence in SCNT embryos. (**a**,**b**) Fluorescence staining pattern and relative fluorescence intensity of H3K18cr in IVF and EFC SCNT embryos (C-SCNT and T-SCNT) at different stages (scale bar, 50 µm). (**c**) The cleavage ratios of embryos at all cleavage stages in IVF, C-SCNT, and T-SCNT. The number of embryos was 40 ± 5 in the IVF, C-SCNT, and T-SCNT groups. C-SCNT (control SCNT) and T-SCNT (experimental SCNT) groups represent cloned embryos generated with untreated and NaCr-treated donor cells (FFCs), respectively. The IVF group represents a blank control group; * *p* < 0.05, ** *p* < 0.01, *** *p* < 0.001; no asterisk indicates no significant difference.

**Table 1 animals-12-02848-t001:** The primer sequences used for qRT-PCR.

Gene Name	Primer Sequence (5′-3′)	Tm (°C)
*GAPDH*	F: TTGTGATGGGCGTGAACC	50
	R: CCCTCCACGATGCCAAA	48
*Nanog*	F: GTCTCTCCTCTTCCTTCCTCCA	59
	R: TCTTCCTTCTCTGTGCTCTCCTC	57
*Oct4*	F: GCCAAGCTCCTAAAGCAGAAGA	62
	R: AAAGCCTCAAAACGGCAGATAG	63
*Sox2*	F: CATGATGGAGACGGAACTGG	55
	R: CGGGCTGTTCTTCTGGTTG	53

## Data Availability

All data are presented in the manuscript.

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
