# Peer review of "Effects of Crotonylation on Reprogramming of Cashmere Goat Somatic Cells with Different Differentiation Degrees"

_animals, 2022, doi:10.3390/ani12202848_

Round 1
Reviewer 1 Report
The present work aims to determine the effect of the modification of the epigenetic status of somatic cell by changing lysine crotonylation on the reprogramming capacity after nuclear transfer. Sodium crotonate is used to modify the lysine crotonylation level in three differentiated cell donors. The approach is interesting and could have a great impact on the improvement of somatic cloning in goats and other species. However, there are several points that the authors should address and improve.
1. line 132, why a concentration of 10mM NaCr selected to determine the optimal incubation time?
2. it seems that Cell viability, Cycle and Apoptosis Analyses are required to determine the effect of the treatment; concentration and incubation but they are presented as independent experiments. Please, explain the rationale of these analyses. Indicate the number of replicates by experimental group for all the analyses.
3. it is suggested including a second housekeeping gene for the normalization of qRT-PCRs and indicate the experimental replicates for each group.
4. Define the maturation medium.
5. Describe activation protocol and reprogramming time. The incubation of reconstructed embryos in SOF during 3.5 hrs is unlikely to be sufficient to activate oocyte metabolism.
6. For each cell line or donor, Indicate the cell passage, the culture conditions for each experiment and the cell confluence for each analysis and at the moment of nuclear transfer.
7. Indicate the age (or a range) of donor of EFCs and sex of all donors.
8. Considering data in figure 3b, it seems that NaCr also induce cell cycle arrest; author should mention the mechanism of action. The protocol to induce cell cycle arrest for cloning should be detailed in materials and methods. This part is confusing and only from the discussion it can be deduced that the G0/G1 rate is used as indicator of toxicity after NaCr treatment. However, the induction of cell cycle arrest is a required step to synchronize cell donor and cytoplast receptor, so a higher G0/G1 rate could be interpretated as a positive sign instead of a negative effect on the cells. Author should be clearer on this aspect and describe the cloning protocol used in this work.
9. Why authors are expecting that the cell donors express the pluripotency markers (OCT4, SOX2 and NANOG)? Use reference to demonstrate that NANOG and SOX2 are expressed in differentiated cells. In the WB figure (Figure 1h) there are several bands for each gene; a molecular weight marker as well as positive and negative controls should be included to confirm the detection of the expected proteins. Discuss how or why the expression of OCT4, SOX2 and NANOG is reduced by the treatment compared to the expression level in differentiated cells (Figure 4).
10. Data regarding embryo production (number of oocytes, IVM rate, number of produced embryos by group, developmental rate, and number of replicates) must be included. Explain why 8 cell stage embryos were considered for the analysis instead of morula and/or blastocysts.
11. There is no analysis on the effect of different cell types on the efficiency of reprogramming after nuclear transfer that is expected considering the use of three different cell lines and the idea that they all have a different potential to be reprogrammed and consequently, to produced cloned embryos. This analysis would be interesting and informative.
12. Author conclude that H3K9cr and H3K18cr modification is similar between clone and IVF embryos, however, the analysis was performed using different cell lines (MDSCs and FFCs: H3K9cr; EFCs: H3K18cr) which weakens the conclusions.
Author Response
Dear reviewers:
Thank you for your letter and for the your comments concerning our manuscript entitled “Effects of crotonylation on reprogramming of Cashmere goat somatic cells with different differentiation degrees” (Submission ID: animals-1890550). This comment is valuable and important guiding significance to our researches. According with your advice, we tried our best to amend the relevant part and made some changes in the manuscript. These changes will not influence the content and framework of the paper. All of your questions were answered below. And here we list the changes and marked in red using the “Track Changes” function in the revised manuscript. Please see the attachment.
Once again, thank you very much for your comments and suggestions.
Yours Sincerely,
Dongjun Liu

Reviewer 2 Report
The manuscript offers a valuable contribution to reprogramming of mammalian somatic cells and SCNT embryos. There are some specific recommendations below for improved clarity and transparency of the various assays that you should incorporate into the methods section. In addition, several important questions the reviewer pointed out need to be explained carefully.
Major revision:
1. Are all follicles from goat ovaries cut for COCs collection? Or specific diameter follicles were used for oocyte collection? What kind of COCs were used for maturation? The COCs derived from various diameter follicles and/or with different morphology (cumulus cells layers) possess different maturation and development capabilities. The reliability of the data needs to be re-considered.
2. You just observed IVF and SCNT embryo cleavage rates. The reviewer was wondering why did you not check the blastocyst formation rate? In addition, the cleavage rate was checked on day 2 of culture?
3. How did you obtain these embryos (1-cell, 2-cell, 4-cell and 8-cell) from Figures 5, 6 and 7? By day of culture or stage? 1-cell, 2-cell, 4-cell and 8-cell stage embryos were collected on day 1, 2, 3, 4 of culture, respectively? Or were all embryos collected on day 2?
4. In the current study, IVF and SCNT embryo cleavage rates were less than 40% and 20%, respectively. Are they normal? Whether the experimental condition is right. Maybe poor experimental conditions led to poor embryo development. For example, there are no blastocysts.
Minor revision:
1. Line 88, page 2. “CO2” should be CO2 (subscript)
2. Line 126, page 3. “0mM10mM” should be “0mM, 10mM”. If possible, please provide NaCr details, for example Catalog Number. In addition, what kind of solvent was used to dissolve NaCr? The NaCr is the most important protagonist in the current study, so the reviewer believes that the authors need to provide more details for NaCr.
3. For cell apoptosis analysis, why do you lose negative or positive control?
4. Please insert information about technical controls (negative or positive) for real-time PCR to verify absence of contamination or genomic DNA.
5. Please provide the annealing temperature for all primer sequences listed in Table 1, it would be helpful to have the information presented in tabular form.
6. You lose negative control in the immunofluorescence analysis section.
7. Line 311, page 10, “qPT-PCR” should be “qRT-PCR”.
8. Please supplement an abbreviation section.
Author Response
Dear reviewer:
Thanks to the editor for giving us an opportunity to modify our manuscript entitled “Effects of crotonylation on reprogramming of Cashmere goat somatic cells with different differentiation degrees” (Submission ID:animals-1890550). We have read all the comments and these comments are valuable and important guiding significance to our researches. We tried our best to revise our manuscript according to the comments. And here we list the changes and marked in red using the “Track Changes” function in the revised manuscript. We have responded to the reviewers’ comments point by point according the comments. please see the attachment.
We appreciate for editor’ and reviewers’ warm work earnestly, and hope that the correction will meet with approval. Should you have any questions, please contact us without hesitate.
Yours sincerely,
Dongjun Liu

Round 2
Reviewer 2 Report
I think all my comments have been adequately addressed. This revised manuscript is improved. A careful proof-reading is required before publishing.